# Effects of the COVID-19 Pandemic on Health Behaviors of Higher Education Students in Ghana: A Cross-Sectional Study

**DOI:** 10.3390/ijerph192416442

**Published:** 2022-12-08

**Authors:** Mary Amoako, Felicity Amoah-Agyei, Gideon Okyere Mensah, Chen Du, Selin Sergin, Jenifer I. Fenton, Robin M. Tucker

**Affiliations:** 1Department of Biochemistry and Biotechnology, Kwame Nkrumah University of Science and Technology, Kumasi 00233, Ghana; 2Department of Food Science and Human Nutrition, Michigan State University, 469 Wilson Rd., East Lansing, MI 48824, USA

**Keywords:** dietary risk, sleep quality, stress, COVID-19, higher education

## Abstract

Stressful events can significantly impact health behaviors of tertiary students in various ways. Many studies reported adverse alterations in health behaviors during the COVID-19 pandemic. However, there is limited knowledge about students from sub-Saharan African countries. Therefore, this study aimed to assess effects of the COVID-19 pandemic on the health behaviors of Ghanaian tertiary students, with an emphasis on the association between sleep and other health behaviors. A cross-sectional study with eligible tertiary students from the Kwame Nkrumah University of Science and Technology in Ghana (*n* = 129) was conducted. An online survey assessed quality and duration of sleep, financial stress, dietary risk, alcohol misuse, and physical exercise using validated tools. Health behaviors did not differ by gender. The COVID-19 pandemic negatively affected health behaviors for many students including dietary intake (20.2%), sleep quality (20.2%) and duration (81.4%), alcohol consumption (3.1%), exercise frequency (4.1%), and exercise intensity (38%). Shorter sleep duration was linked with greater alcohol misuse scores (*p* < 0.05). A majority of the students (56%) had increased financial stress during the pandemic. This study contributes important insights into the effects that stressful events such as a pandemic have on the health of higher education students in sub-Saharan Africa.

## 1. Introduction

The recent COVID-19 pandemic has contributed to detrimental changes in health behaviors and outcomes of university students [1,2,3], but most of the studies to-date have explored the impact of the pandemic on health behaviors in industrially advanced countries [4,5]. Little is known about the influence of the COVID-19 pandemic on sleep and other health behaviors in students living in sub-Saharan African countries, including Ghanaian university students. Measuring these factors in Ghanaian students is important to place their health habits in a global context and address the lack of information on these health behaviors during the pandemic in Ghana.

Sleep is disrupted by stressful events [6]. As outbreaks progressed throughout the world, changes in sleep behaviors coincided with the introduction of remote work and lockdown policies [7]. Numerous studies and press reports indicated that sleep quality and duration was negatively affected during the COVID-19 pandemic [6,8,9,10]. For example, poor sleep quality and increased mental illness were observed in China in younger individuals (<35 years of age) who spent time worrying about the pandemic [6]. In the Netherlands, changes in sleep quality worsened in those who had good sleep before the pandemic, and these changes were related to stress and worry [2]. 

The COVID-19 pandemic also influenced numerous other behaviors. Previous studies documented that in addition to quality and duration of sleep, dietary patterns, alcohol misuse, physical exercise, sitting time, and financial stress were altered by the pandemic in other countries [1,4,5,11]. Importantly, alterations in quality and duration of sleep may be related to the pandemic’s impact on these other health behaviors. In general, poor sleep quality is associated with health complications such as hypertension, obesity, type 2 diabetes mellitus, alcohol abuse, and depression [12,13]. Sleep issues are also associated with increased financial stress [14], less physical activity [15], and poorer dietary choices [16,17]. Understanding whether sleep is associated with other health behavior changes during the pandemic is important to help inform future interventions.

Disruptions in sleep and health behaviors may be worse among students, given that students are at an increased risk for elevated stress and mental health conditions [3,18,19,20]. In surveys including high-, middle-, and low-income countries, students were more likely than non-students to report changes in sleep during the pandemic [21,22]. In addition, students in South Africa had more behavioral disruptions than non-students, reporting greater increases in sitting, delays in sleep time, and insomnia, and larger decreases in exercise and sleep regularity [23]. Disruptions in sleep quality were more likely to be reported by females than males [23]. As health habits are solidified during this stage of life and play a pivotal role in long-term health outcomes, informing public health interventions for students is crucial [24]. 

Therefore, the goal of this study was to characterize health behaviors (quality and duration of sleep, dietary patterns, alcohol misuse, physical exercise, and sitting time) and financial stress of Ghanaian higher education students by gender and to assess how these health behaviors were impacted by the pandemic, with an emphasis on sleep. It was hypothesized that reduced sleep quality and duration during the COVID-19 pandemic would be related to other health behaviors.

## 2. Materials and Methods

### 2.1. Study Design

In this cross-sectional study, undergraduate and postgraduate students from the Kwame Nkrumah University of Science and Technology, Ghana who were at least 18 years old were eligible for enrollment. Students below 18 years of age were excluded. To recruit participants, five departments from each college in the university were randomly selected to be contacted about the study. After contacting course representatives in each selected department, either the course representatives or study personnel distributed the online survey in Qualtrics on students’ WhatsApp platforms. The link was shared with approximately 1000 students. Participants were instructed not to share the survey link on other online platforms. The survey link remained available to students for data collection between October 2020 and January 2021. During this period, in-person learning was halted with online instruction taking place due to the COVID-19 pandemic. A flowchart of the study design is displayed in Figure 1.

The study was approved by the Committee of Human Research Publication and Ethics of Komfo Anokye Teaching Hospital/School of Medical Sciences, Kwame Nkrumah University of Science and Technology Committee on Human Research Publication and Ethics (School of Medical Sciences) CHRPE/AP/389/20 (KNUST). Consent was obtained from all students who agreed to participate in the study. The research was completely anonymous, optional, and participants could withdraw at any time.

### 2.2. Demographics

Age, gender, undergraduate or graduate status, year, and domestic versus international status were obtained. Height and weight of the subjects were self-reported.

### 2.3. Evaluation of Quality and Duration of Sleep

Quality of sleep in the past month was assessed using the validated Pittsburgh Sleep Quality Index (PSQI) questionnaire [25,26,27,28]. Questions about the duration of sleep for weekdays and weekends were asked. Standard duration of sleep for weekdays and weekends were calculated. Differences in duration of sleep between weekdays and weekends were also analyzed.

### 2.4. Evaluation of Dietary Risk and Misuse of Alcohol

The Starting the Conversation (STC) questionnaire was used to assess dietary habits [29]. The STC is a food frequency questionnaire (FFQ) consisting of eight items. The STC evaluated dietary risk scores of participants on the basis of standard eating habits [29]. Parameters such as “weekly consumption of fast-food meals or snacks” and “daily intake of [carbonated sugar-sweetened beverages] and sweet tea” among other important dietary behaviors were assessed. The Alcohol Use Disorders Identification Test Consumption questionnaire (AUDIT-C) measured the misuse of alcohol. The AUDIT-C is a validated tool for screening heavy drinking in the previous year [30]. To estimate the number of drinks consumed by participants, a reference guide with a standard drink equivalents chart was used [31]. On a scale of 0–12 points, alcohol misuse is defined by a score of 4 and greater and a score of 3 and greater for men and women, respectively [32].

### 2.5. Evaluation of Physical Activity and Sitting Time

Evaluation of physical activity levels and sitting time was conducted using the International Physical Activity Questionnaire (IPAQ) for a period of one week retrospectively [33]. Metabolic equivalents (METs) in minutes per week were used to calculate and report total physical activity levels. Physical activity intensity and duration was reported in METs (minutes per week).

### 2.6. Assessment of Financial Stress

The University Student Financial Stress Assessment (USFSA) tool was used to measure financial stress, where a higher score indicates greater stress. This tool has questionnaires contained in the 2010 Ohio Student Financial Wellness Survey [34]. The tool assessed stressful situations linked to an individual’s financial circumstances, such as their ability to finance their monthly expenditures or school fees. 

### 2.7. Evaluation of the Impact of the COVID-19 Pandemic on Health Behaviors 

The impact of the COVID-19 pandemic on dietary intake patterns, intake of alcohol, quality and duration of sleep, and physical activity was assessed as previously described [3]. Example of questions included: Has the COVID-19 pandemic influenced your alcohol intake? Possible answers included: Yes, I have been drinking more; yes, I have been drinking less; no, it has not influenced my alcohol intake; I do not drink alcohol. Has the amount of time you spend exercising changed during the COVID-19 pandemic compared to before the pandemic? Possible answers included: I exercise less, I exercise more, and my exercise level did not change.

### 2.8. Statistical Analysis

Data were cleaned, and validation was performed prior to analysis to ensure data quality. Each variable was screened for outliers. Outliers were defined as values which were three standard deviations (SDs) above or below the mean. With kurtosis and skewness after removal of outliers, all variables were generally normally distributed. Gender differences for age, body mass index, dietary intake risk, misuse of alcohol risk, physical activity, and sitting time were estimated with independent *t*-tests. Chi-square analysis was utilized to test for independence between gender and poor sleep quality/poor sleep duration.

Health behaviors that were affected by the pandemic were compared between male and female students using *z*-tests. PSQI and duration of sleep differences were matched between quality of sleep (worse, better, and no change) and sleep duration (less, more, and no change) using one-way ANOVA. Based on the correction for the false discovery rate, the Bonferroni adjusted *p*-value was used to assess statistical significance at 0.05 alpha level. Linear regression with dummy coding of categorical independent variables was used to ascertain if quality and duration of sleep alterations were associated with dietary intake risk scores, misuse of alcohol scores, physical activity levels, and sitting time. Regression standardized residuals, normal P-P plots, and scatter plots were generated in which normality and homoscedasticity assumptions were met. All analyses were completed using packages in Python 3.8.10 (https://www.python.org/downloads/release/python-3810/, accessed on 1 May 2021), and these packages included Pandas 1.3.0, Numpy 1.3.0, Statsmodels 0.12.2, Scipy 1.7.0, and Scikit-Posthocs 0.6.7.

## 3. Results

### 3.1. Demographic and Anthropometric Parameters

Demographic and anthropometric parameters of students are shown in Table 1. A total of 129 participants were used in analyses after removal of 12 outliers and data with incomplete information. Most of the participants were male (*n* = 72, 55.8%). In terms of college representation, the most responses were collected from the College of Science (*n* = 44, 34.1%), and the least number of responses were recorded from the College of Agriculture and Natural Resources (*n* = 7, 5.4%) (Table 2).

### 3.2. Gender Comparisons of Health Behaviors during the COVID-19 Pandemic

Gender comparisons of each health behavior assessed are shown in Table 3. Quality of sleep during weekdays and weekends did not differ between males and females. The PSQI was used to determine quality of sleep where a higher score designates worsening sleep quality, and a PSQI score > 5 serves as a dichotomous cut-off for poor versus good sleep quality [35]. A similar proportion of females and males reported poor sleep quality (49.1% versus 48.6%) and poor sleep duration (56.1% versus 48.6%), respectively. However, male students had significantly higher financial stress than female students. Dietary risk and alcohol misuse did not differ between males and females. Further, there were no significant differences in physical activity or sitting time by gender; large variability in physical activity and sitting time was measured, reflecting the heterogeneity in METs of the students (Table 3).

### 3.3. Gender Comparisons of COVID-19 Pandemic-Influenced Health Behaviors

Overall, the health behaviors of male and female students were similarly influenced by the COVID-19 pandemic (Table 4). There were no significant differences between male and female students regarding diet, sleep quality and duration, alcohol consumption, exercise frequency, or exercise intensity.

### 3.4. Association between Quality and Duration of Sleep Changes and Health Behaviors

PSQI scores and mean duration of sleep were compared among students who reported worse sleep, better sleep, and no change in quality of sleep during the pandemic. Similarly, these measures were compared among students who reported less sleep, more sleep, and no change in sleep duration during the pandemic (Table 5). A higher PSQI, indicating poorer sleep quality, was observed among students who reported worse sleep compared to students who reported better (*p* < 0.001) and no change (*p* = 0.005) in quality of sleep. Additionally, PSQI scores were similar among students who reported better and no change in sleep quality (*p* > 0.999). Average duration of sleep was highest amongst students who reported sleeping better.

Average sleep duration did not differ between the three groups (those who slept less, those who slept more, and those who had no change in sleep duration) (Table 5). PSQI scores were highest among students who reported less sleep compared to students who reported having more sleep or no change in duration of sleep (*p* = 0.001). PSQI scores were similar between students with more sleep and no difference in sleep duration (*p* = 0.849). While more than half (56%) of the study population reported an increase in financial stress during the pandemic, PSQI scores and average sleep duration did not differ by financial stress status (*p* > 0.05). Further, change in quality and duration of sleep was not associated with dietary risk scores, physical activity levels, or sitting time (Table 6). Less sleep duration was associated with higher alcohol misuse scores; however, change in quality of sleep was not linked with alcohol misuse (Table 6).

## 4. Discussion

The present study characterized and compared health behaviors of students from the Kwame Nkrumah University of Science and Technology, Ghana by gender and assessed how health behaviors were influenced by the COVID-19 pandemic, with an emphasis on the relationship between sleep and other health behaviors. The study reported that health behaviors of males and females were influenced similarly by the COVID-19 pandemic. While less sleep duration was associated with higher alcohol misuse scores, changes in sleep quality and duration during the pandemic were not associated with any other health behaviors. 

Several studies demonstrate that sleep behaviors were altered across the globe during the COVID-19 pandemic [4,6,7,8]. The present study confirmed that the pandemic negatively affected Ghanaian students, both in terms of sleep quality and sleep duration, and that the effects did not differ between males and females. Among Ghanaian medical students before the pandemic, about 56% of students reported poor sleep quality using the PSQI, and this did not vary by gender [36]. Further, 54.1% of a sample of students from the University of Ghana reported poor quality of sleep on the PSQI prior to the pandemic [37]. Poor quality of sleep was associated with metabolic syndrome and obesity [37]. As sleep issues are prevalent among Ghanaian students and sleep can affect many other health outcomes, exploring how measures could be put in place to help students adopt proper sleep behaviors is warranted. 

The present study indicates that both male and female students reported negative changes in dietary patterns. In Ghana, the pandemic severely impacted food availability and accessibility due to rising food prices, the underdevelopment of the food chain, and fear and panic buying [38]. The COVID-19 pandemic also altered how people could access marketplaces in Ghana; the lockdown directive limited people from traveling outside of their neighborhood to find more affordable foods [39]. Among households in Ghana, about 58% reported not having enough to eat during the pandemic, and about 61% reported not having enough income [40]. This experience may differ among university students. A higher number of students from the University for Development Studies in Ghana reported eating homemade meals and choosing healthy foods during the pandemic than before [41]. Despite this positive change, it is well-established that food prices and socio-economic status are related to dietary quality [42,43,44]. Rising food prices, increased financial stress, and limited food availability during the pandemic may have caused some students to purchase less healthy foods in this study. 

There was no observed difference in alcohol misuse scores between male and female students, which is contrary to other reports. In another study of college students from multiple countries, a higher proportion of males reported alcohol misuse compared to females during the COVID-19 pandemic [1]. Various studies suggest that males possess a number of factors that predispose them to disruptive drinking compared to females, including a higher tolerance to alcohol, later brain maturation, perceived peer alcoholism, and their upbringing into traditional gender roles [45,46]. Moreover, among college students, a desire to “fit in” and feelings of loneliness motivate some young adults to engage in unhealthy behaviors including alcohol misuse [47]. Although more male students are expected to engage in alcohol misuse, females may have experienced increased stress levels or greater loneliness that could increase their likelihood to misuse alcohol during the pandemic [48]. However, the proportion of students reporting alcohol misuse in this study was relatively low, so these factors discussed above may not have played a large role.

Over a third of students in this study reported exercising less and performing less intense exercise during the pandemic. As a result of the restrictions imposed on physical activity due to COVID-19, physical activity in Ghanaian adults was reduced [49]. Further, increased sitting and excessive screen time during the pandemic likely contributed to the observed reduction in physical activity [23,49,50]. Reports indicate that men are more likely to exercise in Ghana than women because the representation of strength and universal body image affects male participation in physical activity [51]. However, there were no differences in physical activity by gender in this study, likely because COVID-19 restrictions were imposed on all individuals, irrespective of gender. 

Changes in sleep quality and duration during the pandemic were not associated with most health behaviors assessed in this study. Students who reported either changes in sleep quality or duration had similar dietary risk scores. Previously, poor quality of sleep during the COVID-19 pandemic was associated with poor dietary habits in higher education students in multiple other countries [1]. Further, students and young adults with reduced sleep quality consumed fewer servings of fruits and dairy and increased calories [52,53]. Our findings may be due to changes in food insecurity among Ghanaians during the COVID-19 pandemic, potentially affecting diet quality to a greater extent than sleep [38,54]. Moreover, while less sleep duration was associated with higher alcohol misuse scores, changes in quality of sleep were not associated with misuse of alcohol in the present study. Alcohol abuse is an important concern among students of higher education [55,56,57]. In previous studies, students with reduced quality of sleep consumed alcohol more regularly and in excess and experienced increased undesirable effects including metabolic impairments [58,59,60,61]. The lack of difference in this study may be due to very few students reporting alcohol misuse. In addition, on average, most groups in the analysis had poor sleep quality (PSQI scores). Thus, changes in alcohol misuse scores might not be noticeable. Lastly, changes in sleep quality and duration were not related to physical activity and sitting time. Previously, improved sleep quality and adequate sleep were associated with higher physical activity levels [62,63,64,65]. Further, higher PSQI scores were related to reduced physical activity and more sedentary behaviors in other studies [63,66]. Again, most groups in the present analysis had poor sleep quality; this may have hindered the ability to measure any differences in physical activity.

The authors note that the outcomes of this study cannot be generalized to the entire student population in Ghana given that the study participants were only from the Kwame Nkrumah University of Science and Technology and the sample size was relatively modest. Further, causal relationships between changes in sleep and health behaviors could not be analyzed due to the cross-sectional nature of the study. However, this is one of few studies to assess the effects of the COVID-19 pandemic on various health behaviors in sub-Saharan Africa. Validated survey instruments were used to assess health behaviors of the students, and both undergraduate and postgraduate students were recruited for the study.

## 5. Conclusions

The current study demonstrated that health behaviors of both male and female students were negatively affected by the COVID-19 pandemic; however, no significant differences between males and females were observed. Shorter sleep duration was associated with higher alcohol misuse scores. Altogether, these results indicate that poor sleep quality among students of higher education is a concern among Ghanaian students, especially during the COVID-19 pandemic. More studies investigating the impact of COVID-19 on health behaviors should be conducted in other universities across Ghana to generate adequate evidence-based knowledge to effectively understand the impact of stressful events on students’ health.

## Figures and Tables

**Figure 1 ijerph-19-16442-f001:**
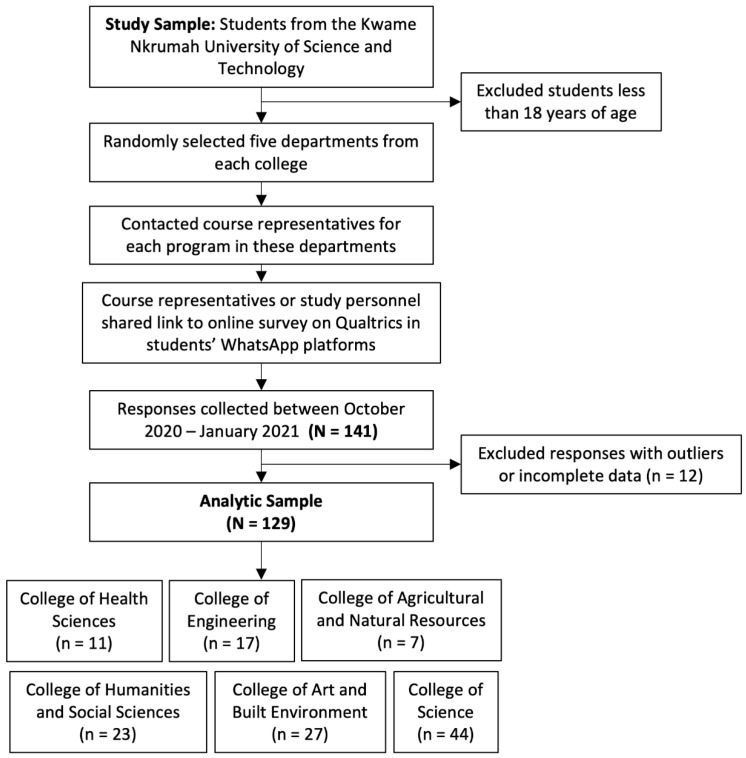
Flowchart of study design.

**Table 1 ijerph-19-16442-t001:** Demographic and anthropometric information of participants (*n* = 129).

Parameters	Males	Females	*p*-Value	t/z-Value	df
*n* (%)	72 (55.8)	57 (44.2)	0.060	1.868	1
Age (years) (mean ± SD)	21.1 ± 1.7	20.9 ± 1.7	0.501	0.674	124
BMI (kg/m^2^) (mean ± SD)	19.0 ± 4.2	19.7 ± 4.6	0.375	−0.891	106

Missing values: BMI, *n* = 13. Independent *t*-tests were used to compare age and BMI of male and female students. The *z*-test showed no difference in the distribution of males and females in the study population, which implies that both genders were equally represented in the study.

**Table 2 ijerph-19-16442-t002:** Distribution of students from the various colleges.

College	% of Students (*n* = 129)
College of Science (CoS)	34.1 (44)
College of Art and Built Environment (CABES)	20.9 (27)
College of Engineering (CoE)	17.8 (23)
College of Humanities and Social Sciences (CoHSS)	13.2 (17)
College of Health Sciences (CoH)	8.5 (11)
College of Agriculture and Natural Resources (CANR)	5.4 (7)

**Table 3 ijerph-19-16442-t003:** Gender comparisons of health behaviors.

	Males	Females	Overall	*p*-Value	t/z/χ^2^-Value	df
PSQI scores (mean ± SD)	5.72 ± 2.67	6.14 ± 3.58	5.91 ± 3.10	0.448	−0.761	126
Average sleep (h/d) (mean ± SD)	6.72 ± 1.32	6.75 ± 1.40	6.73 ± 1.35	0.899	−0.128	123
Weekday sleep (h/d) (mean ± SD)	6.57 ± 1.34	6.68 ± 1.38	6.62 ± 1.35	0.642	−0.465	123
Weekend sleep (h/d) (mean ± SD)	7.16 ± 1.70	6.82 ± 1.89	7.01 ± 1.780	0.289	1.071	125
Poor sleep quality (%)	48.61	49.12	48.84	0.982	0	1
Short sleep duration (%)	48.61	56.14	51.94	0.465	0.532	1
Financial stress (mean ± SD)	16.33 ± 4.68	14.63 ± 4.98	15.58 ± 4.87	0.048	1.993	127
Dietary risk (mean ± SD)	6.24 ± 2.46	6.89 ± 2.72	6.47 ± 2.61	0.153	−1.439	127
Alcohol misuse score (mean ± SD)	0.33 ± 0.65	0.28 ± 0.65	0.31 ± 0.65	0.653	1.417	127
Classified as alcohol misuser (%)	4.20	1.80	0.75	0.432	0.785	1
METs (min/wk)	4022.49 ± 4351.39	3392.79 ± 4597.26	3727.14 ± 4459.45	0.456	0.748	111
Sitting time (min/d)	266.82 ± 138.85	290.5 ± 196.08	277.27 ± 166.23	0.427	−0.796	125

Independent *t*-tests compared PSQI scores, sleep hours, financial stress, dietary risk, alcohol misuse, metabolic equivalents (METs), and sitting time between males and females. Chi-square was used to test for independence between gender and poor sleep quality/poor sleep duration. The difference in proportion of males and females who misused alcohol was tested using proportions *z*-test. PSQI = Pittsburg Sleep Quality Index. PSQI scores range from 0 = best to 21 = worst quality of sleep. PSQI scores > 5 were defined as poor sleep quality. Short sleep duration was interpreted as duration of sleep < 7 h/d (average of weekdays and weekends). Financial stress was measured with the University Student Financial Stress Assessment on a scale of 6–30 points where a higher score indicates higher stress. Dietary risk reflects the score on Starting the Conversation food frequency questionnaire; 0 = best to 16 = worst dietary quality. Misuse of alcohol was determined by score on the Alcohol Use Disorders Identification Test Consumption questionnaire; 0 = no alcohol use to 12 = highest alcohol use. Scores ≥ 3 in females and ≥4 in males were interpreted as a misuser. Physical activity level was defined by METs reported on the International Physical Activity Questionnaire. Missing values: PSQI, *n* = 1; METs, *n* = 15. The adjusted alpha level used is 0.004.

**Table 4 ijerph-19-16442-t004:** Influence of the COVID-19 pandemic on health behaviors.

Health Behavior	Direction of Change	Males (%)	Females (%)	Overall	*p*-Value	z-Value
Diet	Less healthy	19.44	21.05	20.16	0.821	−0.226
Alcohol consumption	Drinking more	4.17	1.75	3.10	0.432	0.785
Sleep quality	Worse	16.67	24.56	20.16	0.267	−1.110
Sleep duration	Less	22.22	22.81	20.48	0.937	−0.079
Exercise frequency	Exercising less	43.06	40.35	41.86	0.757	0.309
Exercise intensity	Less intense	38.89	36.84	37.98	0.812	0.238

Proportions *z*-tests were used to assess differences in variables presented in the table between males and females. Degrees of freedom = 1.

**Table 5 ijerph-19-16442-t005:** PSQI, average sleep duration, average sleep quality and financial stress.

Groups	*n* (%)	PSQI (Mean ± SD)	Average Duration of Sleep (Mean ± SD)
Change in quality of sleep			
Worse	26 (20.16)	8.12 ± 3.18 ^a^	6.40 ± 1.20 ^a^
Better	57 (44.19)	5.11 ± 2.65 ^b,c^	7.03 ± 1.31 ^a^
Did not change	46 (35.66)	5.63 ± 3.03 ^c^	6.55 ± 1.43 ^a^
Change in duration of sleep			
Less	29 (22.48)	8.48 ± 2.9 ^a^	5.95 ± 1.40 ^a^
More	70 (54.26)	4.96 ± 2.64 ^b,c^	7.25 ± 1.17 ^a^
Did not change	30 (23.26)	5.60 ± 2.91 ^c^	6.31 ± 1.22 ^a^

Means having non-identical superscripts were significantly different within the column, *p*-value < 0.05. One-way ANOVA was utilized to compare PSQI and average duration of sleep. PSQI = Pittsburgh Sleep Quality Index.

**Table 6 ijerph-19-16442-t006:** Linear regression models exploring sleep alteration linked to health behaviors.

Predictors	Dietary Risk B (*p*-Value)	Alcohol Misuse Scores B (*p*-Value)	Physical Activity (METs/wk) B (*p*-Value)	Sitting Time (min/d) B (*p*-Value)
R-square	0.06	0.06	0.04	0.03
Constant	10.45 (0.001)	0.52 (0.504)	4103.41 (0.45)	470.94 (0.021)
Change in quality of sleep				
Worse	0.76 (0.327)	−0.17 (0.396)	−740.59 (0.589)	28.20 (0.578)
Better	0.94 (0.116)	0.11 (0.498)	1159.87 (0.281)	−4.14 (0.916)
Did not change (reference)	-	-	-	-
Change in duration of sleep				
Less	−0.85 (0.303)	0.50 (0.021)	672.25 (0.659)	−74.16 (0.173)
More	−0.98 (0.142)	0.14 (0.431)	598.23 (0.625)	−14.61 (0.738)
Did not change (reference)	-	-	-	-
Gender				
Male	-	-	-	-
Female	0.83 (0.077)	−0.15 (0.363)	−316.19 (0.713)	18.64 (0.549)
Age	−0.19 (0.159)	0.02 (0.668)	−54.02 (0.831)	−8.69 (0.347)

The model was designed for each dependent variable, and the models accounted for age and gender. Dietary risk reflects the score on Starting the Conversation food frequency questionnaire; 0 = best to 16 = worst dietary quality. Misuse of alcohol was determined by score on the Alcohol Use Disorders Identification Test Consumption questionnaire; 0 = no alcohol use to 12 = highest alcohol use. Scores ≥ 3 in females and ≥4 in males were interpreted as a misuser. Physical activity level and sitting time were reported on the International Physical Activity Questionnaire. METs, Metabolic equivalents.

## Data Availability

The data presented in this study are available on request from the corresponding author. The data are not publicly available due to ongoing analyses.

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
