# Peer review of "Effects of the COVID-19 Pandemic on Health Behaviors of Higher Education Students in Ghana: A Cross-Sectional Study"

_ijerph, 2022, doi:10.3390/ijerph192416442_

Round 1

Reviewer 1 Report

Dear Authors,

The manuscript entitled “COVID-19 Pandemic’s impact on Health Behaviors of Tertiary Students in Ghana: A cross-sectional study” deals with an interesting and current topic. However, it has serious flaws.

First of all, as it is mentioned among the limitations, the sample is limited to one university; moreover, it is very small, so the results cannot be generalized to the whole population of university students in Ghana, or either to the whole population of the chosen university. It is not clear, what your study population is. Besides, what online platforms were used to gather survey data and how did you ensure that only students from the chosen university answered the questions? Another weakness is that hypotheses are missing, and because of that, it seems your analyses are rather random. The literature review is weak; some sections of the manuscript are over-segmented. The scale of alcohol misuse in section 2.3 is missing. The direction of the significant difference of financial stress between male and female students is not mentioned in the text. Tables 3-5 should be merged. In line 154, you state that “PSQI scores > 5 were defined as poor quality of sleep,” while in line 145 it is stated that “a PSQI score = or > 5 serves as a dichotomous cut-off for poor versus good sleep quality.” Beta values of background variables in the regression model are missing. Strength and weaknesses should not be in a separate section. The Availability Statement is meaningless.

The manuscript is full of grammar mistakes and typos, see, e.g., lines 2, 3, 4, 5, 8, 11, 14, 21, 39, 45, 49, 89, 98, 99, 100, 102, 103, 104, 109, 117, 131, 148, 183, 184, 185, 211, 217, 218, 251, 263, 275, 279, 295, the headers of Tables 3 and 5, and “SSitting” in Table 5. The sentences in lines 146-148, 197-201, and 307-308 should be revised. Formatting of “p” in the text is inconsistent (italic or not).

Reviewer 2 Report

Manuscript ID: ijerph-2000615

COVID-19 Pandemic’s impact on Health Behaviors of Tertiary Students in Ghana: A cross-sectional study

Adjepong, Amoah-Agyei, Mensah, Chen Du, Fenton, Tucker

Broad Comments

The study gives valuable information on health related behavior of university medical students in Ghana during the COVID pandemic. The study is well written, the discussion is balanced, but it needs some improvement in statistical terms. The study should not focus on tertiary students, but on medical students only, because there were no other students included.

Statistical analyses

In Table 1 and Table 4: df, χ2/ t-values should be given.

In Table 3, df and χ2. In Table 5, df and t-values.

In Table 6 df and z-values.  

Table 5: there are huge standard deviations, indicating latent subgroups or at least great heterogeneity in the sample; this could be shortly addressed in the Discussion

Tables in general: authors tend to report values with 1 decimal places in tables and values with 2 decimal places in the text, regularly it would be the other way round

Data presentation in Table 8 does not follow the standards for regression analyses. Why is here a change of analysis rational compared to Table 7? Table 7 is on testing hypotheses of differences between groups, Table 8 as well – which you can tell by the interpretation of Table 8 given in the text. Why then using data analyses for testing correlational hypotheses all of a sudden? It should well be the same analyses (ANOVA) like in Table 7. And this would be more reader friendly.

Specific Comments

Line

Text / remarks / suggestions / proposals

Headline

The sample is very specific (university medical students) so something like “COVID-19 Pandemic’s Impact on Health Behaviors of Medical Students in Ghana: A Cross-Sectional Online Study” may be more appropriate

35

…in developed countries -> industrially advanced countries (?)

75

“soda” -> is not common in all of US or Europe a.o. (instead of “soda” often pop, tonic, fizzy drink, soft drink, means often just “water” in Germany), maybe clarification in a footnote (?)

145 to 150

…“There was no change in PSQI scores between male and female students. A proportion of females than males (48.61%) reported poor sleep quality (49.12% versus 48.61%) and poor sleep duration (56.14% versus 48.61%) respectively. Further, there was a significant difference in financial stress between male and female students (Table 3).”…

Do authors mean:

Both females and males reported no statistically significant differences in poor sleep quality (49.12% versus 48.61%) and poor sleep duration (56.14% versus 48.61%) respectively. However, there was a significant difference in financial stress between male and female students (Table 3).

204 to 209

“While more than half (56%) of the population reported an increase in financial stress during the pandemic, PSQI scores did not differ between decreased financial stress vs. increased financial stress groups (p = 0.998), and decreased financial stress vs. no change in financial stress groups (p = 0.644), or increased financial stress vs. no change in financial stress (p = 0.054).“

Questions: With p = 0.054 there would indeed be a significant difference! Is there a mistake in the p-value?

316

“Validated surveys”-> do authors mean validated instruments, validated measures?

TYPOS

Page / Line

Text / remarks / suggestions / proposals

23-42

Citation, Editor, Copyright – missing

Passim

Blanks between words: 89 (months), 236 (n=0), 184 (less sleep), 185 (sleep), 275 before “Predictors”, 279 before “Further”

Missing lines between tables and paragraphs: after Table 2, after legend to Table 3, after legend to Table 7, 333 before “Author contribution”

Comma in § 2.7 headline

125

„using packages in Python 3.8.10“ -> URL for Python would be nice

170 

“SSitting time” - > Sitting time

319

“study participants were from KNUST only” -> all study participants were from the same university (fun fact: a “knust” by the way, is an apple crunch or a bread crunch in Germany)

Reviewer 3 Report

Dear Authors

the article submitted for review raises an important research topic, especially as the research was carried out in a less wealthy country. The manuscript needs to be corrected and expanded. 

Introduction

Please add in your introduction and refer to other results of the bvs study of students' health behaviour in the era of the Pandemic conducted in less developed countries. The available literature in this area confirms numerous studies on similar groups in other less developed countries.

Materials and Methods

-what on- line platforms were used to collect the data? (line 60)

 -How long was the online questionnaire available to respondents? 

- what were the inclusion and exclusion criteria used in the study? Please complete and elaborate on this in the survey methodology

- I did not see in the research methodology that the research was completely anonymous, voluntary and participants could opt out at any time

- please include in the research material a subsection research procedure where there should be a graph showing the process of conducting the experiment as well as the selection of the research group

- how many people were sent the link with the questionnaire? there is no such information in the manuscript (please include this too in the procedure section)

-what was the return rate of the questionnaires, how many were removed from the database and why?(criteria?)

section 2.7 

please attach in the supplementary materials the original questionnaire (line 99)

Results

Please highlight differences in the tables where the results are presented, even though they were not statistically significant there was a difference in activity between men and women, etc.

Discussion

Please elaborate on the paragraph on the influence of food prices and material status of students on diet. There are a number of publications confirming such a relationship.

 Could the availability of sports facilities have an impact on physical activity levels, how does sedentary time relate to physical activity levels (please also expand on this in your discussion based on the literature)  

Limitations

How was the occurrence of errors in the survey procedure limited, reduced or eliminated?

Round 2

Reviewer 1 Report

The manuscript entitled “Effects of the COVID-19 Pandemic on Health Behaviors of Higher Education Students in Ghana: A Cross-sectional Study” has been developed significantly compared to its previous version. I have only a few minor comments.

Figure 1 and the right side of Table 4 cannot be seen (formatting problems); moreover, the steps of the process shown in Figure 1 should be discussed in details. According to the notes of Table 1 “Z-test showed statistically equal distribution of the study population between males and females” but it is true only at 10% significance level. Items in Table 2 should be in descending order. It is not clear what three groups are mentioned in line 221. When you mention the university, you usually should add the country as well (lines 77 and 240).
